

# Are attractive male crickets better able to pay the costs of an immune challenge?

Clint D. Kelly[1,2], Melissa S.C. Telemeco[2,3] and Lyric C. Bartholomay[4,5]

[1] Département des Sciences Biologiques, Univeristé du Québec à Montréal, Montréal, Québec, Canada
[2] Department of Ecology, Evolution and Organismal Biology, Iowa State University, Ames, IA, United States
[3] Science and Education Division, Pacific Science Center, Seattle, WA, United States
[4] Department of Pathobiological Sciences, University of Wisconsin, Madison, WI, United States
[5] Department of Entomology, Iowa State University, Ames, IA, United States

## ABSTRACT

Reproduction and immunity are fitness-related traits that trade-off with each other. Parasite-mediated theories of sexual selection suggest, however, that higher-quality males should suffer smaller costs to reproduction-related traits and behaviours (e.g., sexual display) from an immune challenge because these males possess more resources with which to deal with the challenge. We used *Gryllus texensis* field crickets to test the prediction that attractive males should better maintain the performance of fitness-related traits (e.g., calling effort) in the face of an immune challenge compared with unattractive males. We found no support for our original predictions. However, that immune activation causes attractive males to significantly increase their calling effort compared with unattractive males suggests that these males might terminally invest in order to compensate for decreased future reproduction.

# INTRODUCTION

Individuals maximize fitness by balancing investment into reproduction with investment into other fitness-related traits, including immunity (*Sheldon & Verhulst, 1996*; *Viney, Riley & Buchanan, 2005*; *Schmid-Hempel, 2011*; *Jacobs & Zuk, 2011*). Conflict between competing fitness-related traits means that allocation of resources toward immune function by an infected individual will reduce the resources available for other fitness-related traits, such as sexual display, and vice versa (*McKean & Nunney, 2001*; *Jacot, Scheuber & Brinkhof, 2004*; *McKean & Nunney, 2005*; *Leman et al., 2009*; *López, Gabirot & Martin, 2009*). Individuals will always face trade-offs when partitioning resources among competing functions (*Schmid-Hempel, 2011*; *Jacobs & Zuk, 2011*). However, the relationship between competing functions can be either positive or negative when viewed across individuals. If individuals genetically vary more in their resource allocation than their genetic variation in resource acquisition (*Van Noordwijk & De Jong, 1986*), a trade-off will emerge among individuals because those that allocate more resources to sexual signals will have fewer resources available for

Corresponding author
Clint D. Kelly, kelly.clint@uqam.ca, clintdkelly@icloud.com

immunity (*Schmid-Hempel, 2011*; *Jacobs & Zuk, 2011*). For example, *Skarstein & Folstad (1996)* revealed that male Arctic charr (*Salvelinus alpinus*) with more-colourful sexual ornamentation had weaker immune function.

In contrast, if individuals genetically vary more in their ability to acquire resources than in their allocation of them, then positive genetic covariance between immune function and signal quality will emerge. Empirical evidence indicates that holding access to resources constant results in some individuals producing higher quality sexual signals than others (*Wilkinson, Presgraves & Crymes, 1998*; *Bortolotti et al., 2006*; *Karu, Saks & Hõrak, 2007*; *Thomson, Darveau & Bertram, 2014*). Parasite-mediated theories of sexual selection (*Schmid-Hempel, 2011*; *Jacobs & Zuk, 2011*) hypothesize that positive covariation between immune function and signal quality allow females to acquire indirect genetic benefits for their offspring through mate choice. A positive relationship between immunity and a sexondary sexual character has been demonstrated in several empirical studies (*Saino & Møller, 1996*; *López, 1998*; *Mougeot & Redpath, 2004*; *Kelly & Jennions, 2009*; *Schmid-Hempel, 2011*; *Jacobs & Zuk, 2011*). For example, *Mougeot & Redpath (2004)* found that the quality of a sexual ornament (redness of beak and eye rings) in male red-legged partridge (*Alectoris rufa*) positively correlated with greater swelling response to PHA.

A third pattern has also commonly been reported in the literature wherein individuals increase their investment in a fitness-related trait after an immune challenge. The 'reproductive compensation hypothesis' posits that immune-challenged individuals adaptively shift more of their resources into the current reproductive event because their residual reproductive value is reduced (*Minchella & Loverde, 1981*; see also the terminal investment hypothesis: *Clutton-Brock, 1984*). This hypothesis has been supported empirically in several studies (*McCurdy, Forbes & Boates, 2000*; *Agnew, Koella & Michalakis, 2000*; *Reaney & Knell, 2010*; *Kivleniece et al., 2010*; *Krams et al., 2011*; *Nielsen & Holman, 2012*; but see *Kolluru, Zuk & Chappell, 2002*; *Vainikka et al., 2007*) with, for example, *Polak & Starmer (1998)* showing that male *Drosophila nigrospiracula* parasitized with mites court females at a significantly higher rate than unparasitized males. Although we know that an immune-challenge can increase reproductive investment, we know very little about how individual quality or sexual attractivenss mediates this effect.

Here, we investigate whether sexual attractiveness affects the performance of a sexual signal (calling frequency) after an immune challenge in the Texas field cricket, *Gryllus texensis* (Orthoptera: Gryllidae). Mate choice studies have shown that sexually attractive male crickets can have a particular pheromone profile (*Tregenza & Wedell, 1997*), larger body size (*Simmons, 1986b*; *Bertram & Rook, 2012*), call more frequently (*Hunt et al., 2004*), or have a calling song with, for example, a higher chirp rate, longer intercall duration or louder amplitude (*Wagner & Hoback, 1999*; *Holzer, Jacot & Brinkhof, 2003*; *Scheuber, Jacot & Brinkhof, 2003*; *Brooks et al., 2005*). Although most mate choice studies tend to examine only a small subset of male traits, it is likely that female crickets simultaneously assess several traits when making their mate choice (*Simmons, 1986a*; *Shackleton, Jennions & Hunt, 2005*; *Bussiere et al., 2006*). The above-listed male sexual signals, particularly those related to calling, are condition-dependent and energetically costly to produce. Consequently, the quality of the signal tends to suffer when males are forced to allocate

resources to other fitness-related traits. For example, *Jacot, Scheuber & Brinkhof (2004)* showed that an immune challenge with lipopolysaccharide (LPS) causes a significant reduction in calling rate in male *G. campestris* and *Fedorka & Mousseau (2007)* observed a post-challenge increase in interpulse interval in the calling song of male *Allonemobius socius* ground crickets.

We used controlled laboratory experiments to test the sexual selection hypothesis that sexually attractive males are better able to bear the costs of an immune challenge and thus maintain the performance of one component of sexual signalling, that is calling frequency, during immune activation. We predicted that immune-challenged attractive males would not suffer a significant decrease in calling frequency compared with control males, whereas the calling effort of unattractive males would significantly decline relative to controls after an immune challenge. Alternatively, if sexual signaling trades off with immunity, we would expect that attractive males would suffer a significant drop in signalling relative to controls while unattractive males would not. If immune-challenged males undergo reproductive compensation then attractive and unattractive males should both elevate their sexual signalling relative to controls.

## MATERIALS AND METHODS

Experimental crickets were lab-reared descendants of individuals originally caught in Austin, TX (USA) in 2012 and 2013. The laboratory colony of crickets was reared in several large communal plastic bins (73 × 41 × 46 cm) until their penultimate instar at which time they were transferred to large sex-specific communal bins to prevent mating and ensure virginity. Newly eclosed adults were placed in individual 10 cm deli cups. All crickets were housed in an environmentally controlled room (27 °C, 12:12 h light:dark cycle, 80 % relative humidity) and were supplied with cotton-plugged water vials and dry cat food (Special Kitty Premium Cat Food) *ad libitum*. Crickets were used in experiments 10–14 d post-eclosion to ensure sexual maturity.

### Quantifying attractiveness

As in most gryllid species, mating in *G. texensis* follows a highly stereotypical sequence of behaviours. Males will contact a female with his antennae and then he will produce a courtship call during which he moves backward toward the female. If the female mounts the male he will attempt to attach an externally positioned spermatophore. Spermatophore transfer is accompanied by rapid and irregular flicking of the male's caudal cerci and takes 5–6 s, immediately after which the male unhooks his genitalia. Mating lasts 3 min on average and requires the active cooperation of the female to be successful.

Latency to mate is a reliable predictor of male sexual attractiveness and mating success in field cricket species (*Simmons, 1987a*; *Bateman, 1998*; *Shackleton, Jennions & Hunt, 2005*; *Bussiere et al., 2006*). Following *Shackleton, Jennions & Hunt (2005)* we determined male attractiveness by conducting a four-round no-choice tournament that indexed male attractiveness based on the time that elapsed until a female mounted them. Studies have traditionally used a single trait to assess male attractiveness in crickets (*Heisler, 1985*; *Wedell & Tregenza, 1999*) but a no-choice tournament is a superior approach

because it simultaneously incorporates all relevant factors contributing to short-range male attractiveness (*Head et al., 2005*; *Shackleton, Jennions & Hunt, 2005*; *Bussiere et al., 2006*). For example, several studies on crickets have shown that male attractiveness is unrelated to body size or measures of body condition only (e.g., *Simmons, 1987b*; *Gray & Eckhardt, 2001*; *Shackleton, Jennions & Hunt, 2005*). Tournaments commenced at the onset of the environmental chamber's dark cycle and were conducted under red light to minimize observer disturbance. In the first round, we placed each of 12 sexually naive males in individual plastic containers (10 cm diameter) with a randomly assigned virgin female from our stock culture and observed them until the female mounted the male. We scored a mounting as successful if (1) the female remained motionless on top of the male for at least 3 s and (2) the male commenced spermatophore transfer, characterized by the vibration of his cerci. Pairs were separated prior to spermatophore transfer. Males were ranked 1–12 in order of mounting (rank 1 being the fastest) and males that remained unmounted after 60 min were given the average of the remaining ranks. This process continued for three more rounds with a new female being assigned to each male in each round. Ranks from the four rounds were summed for each male (four lowest sums were attractive; four highest sums were unattractive). Seven such tournaments were completed yielding 28 attractive males (four lowest-ranked males from each tournament) and 28 unattractive males (four highest-ranked males from each tournament).

## Administering an immune challenge

We immune-challenged the attractive and unattractive males the day after the no-choice tournaments by following protocols established for *G. texensis* (*Adamo, 1999*). Briefly, we cold-anesthetized males by placing them on ice for 10 min and then injected them by inserting a pulled-glass microcapillary needle (needles were used only once) along the left pleural region of their abdomen. Microcapillary needles were made in a Flaming/Brown Micropipette Puller (Sutter Instrument Co. model P-97, program 27; Sutter Instrument Co., Novato, CA, USA) with Kwik–Fil's borosilicate glass capillaries. Fourteen attractive males and 14 unattractive males were injected with 5 µL of saline (phosphate-buffered saline, Sigma-Aldrich), and 14 attractive males and 14 unattractive males were injected with 100 µg of lipopolysaccharide from *Serratia marcescens* (LPS, Sigma-Aldrich, St. Louis, MO, USA), dissolved in 5 µL of saline. *S. marcescens* is a soil microbe that is frequently used as an immune challenge model in *G. texensis* because it co-ocurrs with this cricket species in nature and is lethal to it (*Adamo, Jensen & Younger, 2001*). LPS is a non-pathogenic and non-living elicitor that stimulates several pathways in the immune system of orthopterans (*Jacot, Scheuber & Brinkhof, 2004*; *Fedorka & Mousseau, 2007*; *Leman et al., 2009*) including *G. texensis* (*Adamo, 1999*).

## Quantifying sexual signaling

One hour after males were injected (and at the onset of the dark cycle) they were put into a 230 mm ×155 mm ×170 mm plastic arena (Exo Terra Faunarium; Rolf C. Hagen Inc., Montreal, Canada). We began recording calling frequency after 1 h because an immune challenge produces measurable physiological effects in *G. texensis* after 90 min (*Adamo et*

*al., 2008*); this timeframe, therefore, ensured that we captured the full time period. The ventilated section of the lids were removed and replaced with a black mesh screen. The arena contained a paper shelter (made from an Oxford $12.7 \times 7.6$ cm index card cut into 4 strips), one piece of Special Kitty cat chow, and a water vial affixed with hot glue to white paper lining the bottom. During each trial a microphone (Dynex USB, DX-USBMIC13; Dynex, Richfield, MN, USA) was directed toward each male for 5 s every five minutes to assess male calling. For each male, the microphone was held close to the mesh on the top of its arena (within 23 cm of the male). We used QuickTime Player (version 10.2, Apple Inc.) to visualize sounds detected by the microphone and recorded whether or not an individual was calling (0 = no, 1 = yes) during each of the 60 5 s sample periods. The trials were conducted in a dark room but to assist in appropriately placing the microphone near the focal male, the arenas were illuminated with four CMVision IR200–940 (18 W) infra-red Illuminators and visualized with a Canon Vixia HFG10 HD camcorder. Trials were 5 h in duration and so each male's calling was sampled for a total of 5 min over the course of 5 h (i.e., sampled for 5 s every 5 min for 5 h). Trial duration was within the window of immune-activation for LPS. Although LPS is cleared from insect haemolymph within hours (*Kato et al., 1994*) it induces a prolonged up-regulation of some immune parameters in orthopterans (i.e., for a period of days to weeks: *Jacot et al., 2005*; *Fedorka & Mousseau, 2007*; *Kelly, 2011*; but see: *Adamo, 2004*).

## Morphological traits

Immediately prior to being placed in a trial, males were weighed on a Denver Instruments TP-64 digital balance (to the nearest 0.01 g) and the pronotum length measured using a stereoscope equipped with Leica LAS image analysis software (Leica Microsystems Inc., Buffalo Grove, IL, USA). Pronotum length (the distance from the anterior to posterior edges of the pronotum at the midline) is an excellent proxy for body size in *G. texensis* (*Kelly, Tawes & Worthington, 2014*).

## Statistical analysis

The ability of males to bear the cost of an immune-challenge might be related to their body size or body mass (scaled to body size), and not to their attractiveness *per se*, because a larger body can potentially house a greater volume of hemolymph and immune-response substances (*Zuk et al., 2004*; *Fedorka, Zuk & Mousseau, 2004*; *Rantala & Roff, 2006*; *Kelly & Jennions, 2009*). Therefore, we assessed whether body size or scaled mass differed between attractiveness groups and thus should be statistically controlled. We examined whether attractive males differed phenotypically from unattractive males by using one-way analysis of variance (ANOVA) to compare pronotum lengths (proxy for body size) and scaled mass indices (proxy for body condition). Body condition was calculated using the scaled mass index (SMI) following *Kelly, Tawes & Worthington (2014)*. We forgot to weigh and measure one unattractive and one attractive male prior to the experiment so the sample sizes differ between the morphological ($N = 54$) and calling ($N = 56$) analyses. A generalized linear model was used to test whether the fixed factors male attractiveness (attractive or unattractive), immune status (LPS- or saline-injected), and time since injection affected

**Table 1  Results for statistical models examining effect of experimental treatment on and male sexual attractiveness on calling effort.** Results from models (see text) with $Z$ tests for estimated parameters. Values in bold are statistically significant at alpha = 0.05.

| Response | Predictor | β | SE | Z | P |
|---|---|---|---|---|---|
| (A) Calling ($N = 56$) | Intercept | −1.911 | 0.652 | −2.931 | 0.003 |
| | Attractiveness (Un) | −1.855 | 0.984 | −1.886 | 0.059 |
| | **Immune status (Sa)** | **−2.056** | **1.012** | **−2.033** | **0.042** |
| | **Time** | **0.004** | **0.001** | **3.205** | **0.001** |
| | **Immune status (Sa): Attractiveness (Un)** | **3.224** | **1.425** | **2.263** | **0.024** |
| | Immune status (Sa): Time | 0.003 | 0.002 | 1.516 | 0.130 |
| | Attractiveness (Un): Time | 0.002 | 0.002 | 1.374 | 0.170 |
| | Immune status (Sa): Attractiveness (Un): Time | −0.004 | 0.003 | −1.419 | 0.156 |
| (B) Attractive males ($N = 28$) | Intercept | −1.773 | 0.563 | −3.15 | 0.001 |
| | **Immune status (Sa)** | **−1.942** | **0.881** | **−2.204** | **0.027** |
| | **Time** | **0.003** | **0.001** | **3.205** | **0.001** |
| | Immune status (Sa): Time | 0.003 | 0.002 | 1.516 | 0.129 |
| (C) Unattractive males ($N = 28$) | Intercept | −4.074 | 0.952 | −4.278 | $1.88 \times 10^{-05}$ |
| | Immune status (Sa) | 1.298 | 1.152 | 1.127 | 0.260 |
| | **Time** | **0.005** | **0.001** | **4.392** | **$1.12 \times 10^{-05}$** |
| | Immune status (Sa): Time | −0.000 | 0.001 | −0.378 | 0.705 |

the number of calls (family = Poisson). We pooled calling data into ten 30 min bins for analysis. All statistical analyses were conducted in *R3.2.3* (*R Development Core Team, 2015*) with data visualized using *ggplot2* (*Wickham, 2009*). Means are given ±1 SE and $\alpha = 0.05$.

## RESULTS

Attractive and unattractive males did not differ significantly in either pronotum length (one-way ANOVA: $F_{1,52} = 0.001$ $p = 0.98$) or scaled body mass prior to experimental treatment (one-way ANOVA: $F_{1,52} = 0.361$ $p = 0.55$).

We found a significant interaction between male attractiveness and immune status on calling frequency (Table 1A and Fig. 1). We explored this interaction further by examining the effect of immune status on calling frequency separately within each attractiveness category. These analyses found that attractive males called more frequently on average when injected with LPS ($8.36 \pm 2.74$ calls/5 min, $n = 14$) than with saline ($3.28 \pm 1.54$, $n = 14$) (Table 1B) whereas LPS-injected ($6.14 \pm 3.02$, $n = 14$) and saline-injected ($8.64 \pm 3.39$, $n = 14$) unattractive males did not differ in their calling effort (Table 1C). Multiple comparisons using Holm's method showed that saline-injected attractive males did not differ significantly from unattractive males that were either saline-injected ($z = -1.913$, $p = 0.223$) or LPS-injected ($z = 0.698$, $p = 0.485$). All males called more frequently as trials progressed (Table 1).

## DISCUSSION

We found that an immune challenge had little statistical effect on the calling effort of unattractive males whereas an immune challenge caused attractive males to *increase* their calling frequency compared with saline-injected controls. This result does not support the

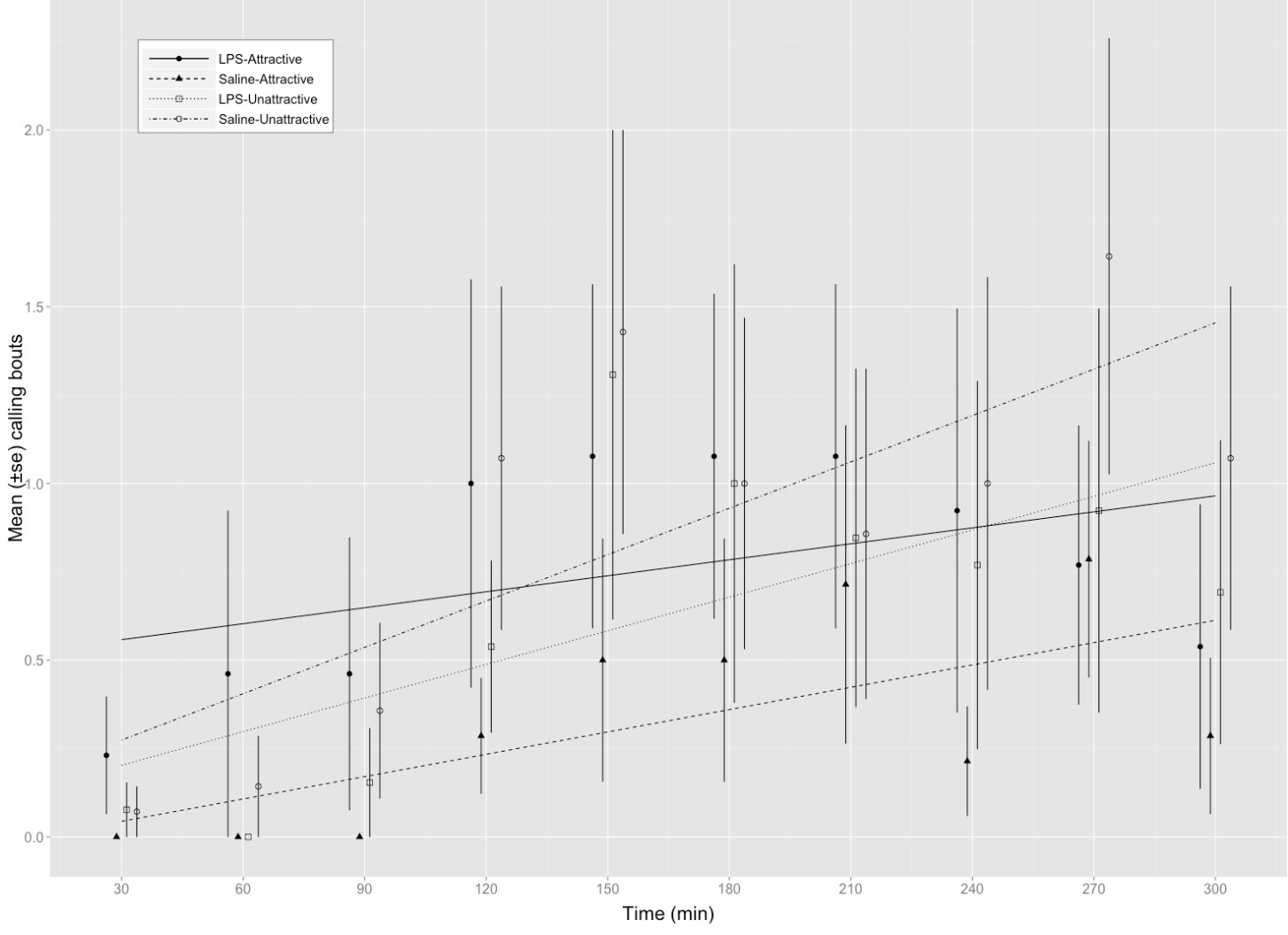

**Figure 1  Calling bouts for attractive and unattractive males that were injected with either saline or LPS.** Mean (±SE) number of calls made by male *G. texensis* crickets during each of the ten 30 min observation periods. Regression lines for each factorial combination were fit using a mixed model (poisson family of errors) with male ID entered as a random effect. LPS-A, LPS-injected attractive males (*n* = 14); Saline-A, saline-injected attractive males (*n* = 14); LPS-U, LPS-injected unattractive males (*n* = 14); Saline-U, saline-injected unattractive males (*n* = 14).

sexual selection hypothesis that the calling effort of unattractive males should decline after an immune challenge while that of attractive males would not.

In contrast, that immune-challenged attractive males elevated their calling rate relative to controls supports the reproductive compensation hypothesis (see also the terminal investment hypothesis: *Clutton-Brock, 1984*): by increasing invesment in current fitness attractive males might compensate for their reduced future fitness. On the other hand, the lack of difference between control and treatment unattractive males does not support the reproductive compensation hypothesis. The observed similarity between saline- and LPS-injected unattractive males is counter-intuitive because immune-challenged unattractive males should have the poorest prospects for survival and, thus, the most to gain from increased investment in reproduction. Our results raise the question as to why attractive,

but not unattractive, males were able to increase their calling effort post-challenge. Perhaps attractive males possessed a larger pool of resources from which to draw for use in both signalling and immunity compared with unattractive males. The scaled mass indices did not differ, on average, between the male attractiveness groups and suggests that the males in both groups had similar energetic reserves (i.e., fat load) and water content since these two variables scale positively with SMI (see *Kelly, Tawes & Worthington, 2014*; *Gray & Eckhardt, 2001*). Despite the similarity in these two components of scaled mass there might still have been important differences between the two attractiveness groups in their resource pools or the capacity to efficiently use their resources (*Hill, 2011*). For example, attractive males might have had more of a certain micronutrient that is critical to immune function or had better enzyme activity (see *Thomson, Darveau & Bertram, 2014*). Thus, perhaps immune-challenged unattractive males *are* terminally investing, but since they have a poor resource pool from which to draw (or inefficient physiological mechanisms), that after allocating some portion of resources to immunity they can only achieve calling rates on par with healthy counterparts (i.e., they cannot increase calling rates relative to control males).

Alternatively, perhaps unattractive male *G. texensis* invest in sexual signaling at the expense of survival (i.e., immunity) regardless of whether they are sick or healthy. In other words, unattractive males might adopt a strategy that resembles reproductive compensation (and terminal investment) simply given their poor prospects of acquiring mates and whether they recieve an immune-challenge makes little difference to their resource allocation strategy. This 'live fast, die young' hypothesis predicts that there should be little reproductive benefit to holding back resource consumption (e.g., via a reduced signaling rate) if caution yields no mates, and thus, no fitness gains. That unattractive males (both saline- and LPS-injected) in our study appear to call at a rate similar to that of healthy attractive males suggests that unattractive males are investing relatively heavily in signaling. This is likely not a general pattern across taxa as *Hunt et al. (2004)* found that it is high-quality, attractive male *T. commodus* crickets that invest in calling at the expense of longevity.

Both alternative explanations would benefit from an examinination of the investment in immunity by healthy and sick individuals with the prediction being that terminally-investing males should have a weaker immune response due to a shift of resources into current reproduction. However, *Sadd et al. (2006)* showed that immune-challenged males that were apparently terminally investing in current reproduction also had significantly higher phenoloxidase activity compared with unchallenged males.

We also found that time had a significant effect on calling effort with all males calling significantly more often as trials progressed over the course of our 5 h observation period. The lack of a significant treatment by time interaction suggests that males increase their calling effort independent of immune status or sexual attractiveness. *Rost & Honegger (1987)* also observed that *G. campestris* males in the wild increase their calling during the period from just after sunset to midnight. Similarly, *Bertram, Xochitl Orozco & Bellani (2004)* showed that wild-caught male *G. texensis* increase their rate of calling throughout the night and posited that this phenomena is an adaptive strategy because there is a

greater abundance of acoustically-orienting parasitoid flies early in the evening and more abundant females later in the evening (*Bertram, Xochitl Orozco & Bellani, 2004*). However, *Bertram (2002)* also showed that lab-reared males tend to call all night with only a small proportion calling more as night progresses. This latter study therefore suggests that perhaps the increased rate of calling that we observed throughout the night in our lab-reared population is due to crickets generally overcoming injection-related injuries.

Our study tested the hypothesis that sexually attractive (i.e., higher-quality) males suffer smaller costs to fitness-related traits from an immune challenge because these males possess more resources with which to pay such costs. We found that attractive males significantly increased their calling effort after an immune-challenge while an immune challenge had little effect on signaling in unattractive males. We suggest that our results might be explained by the reproductive compensation hypothesis but more testing is required to unequivocally support this conclusion.

## ACKNOWLEDGEMENTS

We thank Sue Bertram, Felipe Dargent, and an anonymous reviewer for comments on a previous version of the manuscript.

### Funding

This study was supported by a faculty start-up grant from Iowa State University and a Discovery Grant from the Natural Sciences and Engineering Research Council of Canada to CDK. The funders had no role in study design, data collection and analysis, decision to publish, or preparation of the manuscript.

### Grant Disclosures

The following grant information was disclosed by the authors:
Iowa State University.
Natural Sciences and Engineering Research Council of Canada.

### Competing Interests

The authors declare there are no competing interests.

### Author Contributions

- Clint D. Kelly conceived and designed the experiments, analyzed the data, contributed reagents/materials/analysis tools, wrote the paper, prepared figures and/or tables, reviewed drafts of the paper.
- Melissa S.C. Telemeco conceived and designed the experiments, performed the experiments, analyzed the data, reviewed drafts of the paper.
- Lyric C. Bartholomay conceived and designed the experiments, reviewed drafts of the paper.

## Data Availability

Data available from the Dryad Digital Repository: http://dx.doi.org/10.5061/dryad.tm508 .

## Supplemental Information

Supplemental information for this article can be found online at http://dx.doi.org/10.7717/peerj.1501#supplemental-information.

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
