# Peer review of "Are attractive male crickets better able to pay the costs of an immune challenge?"

_PeerJ, doi:10.7717/peerj.1501_

## Round 0.1 · original submission · Minor Revisions

· Academic Editor

Minor Revisions

Overall this was an extremely well written manuscript that was a pleasure to read. I particularly liked how you ascertained attractive versus unattractive. Using a round-robin tournament where you obtain four scores of mounting latency is a beautiful way to capture all the nuances that go into mate choice. Typically, one would look solely at body size, signalling time, or signal quality, which would eliminate so many of the other components that influence mate choice. Nicely done.

Please pay close attention to the reviewers comments: Reviewer 1 - honest signalling, Figure 1, 5 hrs of recording issue, why is size not significant, temporal variation in calling effort, treatment vs manipulation, and alternative theories; Reviewer 2 - problems with addressing second hypothesis (variation in acquisition), LQ vs HQ condition.

The following publications may also be useful to you in dealing with these revisions, especially explaining the differences in signalling effort related to body stores in the discussion (Thomson et al 2014) and temporal signalling patterns (Bertram et al 2004; Bertram 2002). Please note I am NOT trying to get you to cite my work. In fact, I struggled whether to inform you of them at all, given the conflict of interest as I am your editor. I opted to inform you of them because I felt that they may help to strengthen your argument. Whether you add these publications or not will NOT influence my editorial decision.

Thomson, I.R. C.-A. Darveau, and S.M. Bertram. 2014. Body morphology, energy stores, and muscle enzyme activity explain cricket acoustic mate attraction signalling variation. PLOS ONE (Accepted Feb 3 2014 PONE-D-13-49243R1)

Bertram, S.M., S.X. Orozco, and R. Bellani. 2004. Temporal shifts in conspicuousness: mate attraction displays of the Texas field cricket, Gryllus texensis. Ethology 110(12):963-975.

Bertram, S.M. 2002. The influence of rearing and monitoring environment on temporal mate signaling patterns in the field cricket, Gryllus texensis. Journal of Insect Behavior 15(1):127-137.

Lastly, I noticed a few typos (over and above the issues raised by Reviewer 2), so please read the revised manuscript extremely carefully prior to resubmission.

Reviewer 1 ·

Basic reporting

The authors do a good job providing a thorough yet succinct introduction to the manuscript. Lines 55/56 repeat information already in the first paragraph and can be removed. In line 58 the authors refer to a trade-off 'among individuals', it is unclear to me what this means in contrast to life-history/evolutionary trade-offs. One body of work that seems to be relevant here but the authors neglect is honest signalling. Indeed, if we are to expect that more attractive males are signalling their quality, it is a safe assumption that these males should also be better capable of mitigating the cost of an immune-challenge. Introducing this idea in the introduction, albeit briefly, may allow for a more robust interpretation of the results. At several points throughout the manuscript there are either awkward sentences or overt errors in script (e.g. lines 74;127;158), a thorough edit should address these.

Figure 1 is not entirely clear. If each square on the plot (broken up into quadrants) represents 30 minutes along the x-axis and 0.5 calling bouts along the y axis, how are there data points both before the 30 minute marker on the x-axis (presumably taken between minutes 15 and 30) and after the 300 minute marker (presumably taken between minutes 300 and 315). Finally, the legend is difficult to read, perhaps it can be incorporated in the caption or made to be bigger in the figure (or both).

Both the hypothesis and objective of this study were clear. The methods used to determine attractive and unattractive males were interesting and might be likely to capture a more ‘holistic’ estimate of mate choice due to the variety of traits that females use when assessing males. However, it is unfortunate that the authors do not attempt to verify their assessment of male attraction with their own data (e.g. male size). Size is known, as noted by the authors, to be an attractive trait in males, and though the size data are presented (lines 214-216), the authors do not address the fact that their 'attractive' males did not differ from unattractive males with respect to size. It seems useful to validate their novel approach to determining attractive males by comparing their results to a well-known estimate of male attractiveness (size). The fact that they did not find this association is worthy of discussion.

Experimental design

The method used to assess calling behavior is my main concern with this manuscript. That males were only measured for 5 hours immediately following infection is concerning. First, it is known (and the authors do note) that the effects of an immune response can persist for days, thus making this estimate merely a snapshot of what may be happening with these males.

Second, male calling behavior varies through the night, with some males calling predominantly at dusk and others increasing calling later in the evening, thus, the measure here used cannot account for differences in timing of calling throughout the night and are biased towards males that call at the specific timing in the night cycle during which these measures were taken. Additionally, the authors do not mention when in the day/night cycle calling behavior was measured in these males, increasing the difficulty in determining what impact male call timing could have on the results.

Finally, perhaps a result of both the short period of time measured as well as the immediacy with which it followed injection, it would be nice to see data that suggests the observed calling behavior was resulting from the treatment rather than the manipulation. That is to say, the observation that saline and immune challenged males did not differ in calling, as well as all treatments increasing calling activity as the night progressed (Fig.1) could also be explained by males being generally 'disturbed' by the handling that occurred during infections, and then regained confidence and increased calling the more time that elapsed since injection.

Validity of the findings

The conclusions are sound, and the authors do an excellent job (once again) in providing a succinct yet thoughtful summary of the findings. However, the authors could incorporate alternative theories that their data also address (i.e. honest signalling as mentioned above). Finally, there seems to be greater need to discuss some of the less obvious results such as the lack of difference between control 'attractive' and 'unattractive' males. This would be especially useful given the novel methods used to determine A and U males, methods that did not seem to be supported by their morphometric data.

·

Basic reporting

No Comments

Experimental design

The submission meets the journal Experimental Design criteria. One point should perhaps be addressed in more detail by the authors:
The choice of calling frequency as the a costly signalling trait seems adequate a priori - yet the lack of differences in this trait among attractive control males, unattractive control males and unattractive immune-challenged males could indicate that the trait is not costly, which would have implications for the interpretation of the results. Addressing this point in more detail would improve the manuscript.

Validity of the findings

In general the article meets the journal criteria. At points the discussion is highly speculative, yet the interpretations are well argued for. Speculation is, nonetheless, stated as such.

Additional comments

The authors use male crickets (Gryllus texensis) to test three competing hypotheses about the relationship between immunity and sexually selected traits. The first two hypotheses focus on whether investment in immunity and sexual-traits is primarily influenced by variation in either allocation or acquisition of resources (i.e. van Noordwijk and De Joong 1986, Am Nat), while the third hypothesis focus on terminal compensation. The authors find that high quality (HQ) males challenged by lipopolysaccharides from a soil bacteria (Serratia marcescens) show higher calling frequency (a costly secondary sexual character) than control HQ males (saline injected), and take this result as evidence of terminal investment. In contrast low quality (LQ) males show no difference in calling frequency between challenged and control males, which the authors argue is caused by resource limitation (to allow for terminal investment). Furthermore, LQ males are not significantly different from HQ control males; for this finding the authors attempt and intelligent and plausible, although highly speculative, explanation. Overall, I find the manuscript well written and much interesting. In particular, the authors do an outstanding job at outlining in a clear manner the relationships between resources, immunity and sexual signals. Below I highlight a few issues which, if addressed, could help improve the manuscript.

Major comments
1) Although the framework of three competing hypotheses proposed by the authors works well and is engaging, I do not think that it is possible to address the second hypothesis (variation in acquisition) given their experimental design: crickets were fed ad libitum and therefore there is little room for differences in acquisition (unless individuals had unequal access to the food). This is further supported by the lack of differences in condition (i.e. “scaled body mass”) between HQ and LQ males.
2) The fact that HQ and LQ males did not differ in condition (i.e. SBM) might be problematic. If calling frequency is a “condition-dependent” trait (line 96) and condition does not differ between the two “mate-quality” groups (HQ vs. LQ) it is not surprising that calling frequency did not differ for control HQ, control LQ and challenged LQ (lines 261-263); and “mate quality”/ attractiveness might have been determined by traits that were independent of acquisition and allocation differences (e.g. male pheromone profile). If this is the case, predictions about how male attractiveness affects calling rate after an immune challenge become less straightforward. Could it also have been the case that calling rate did not contribute much to male attractiveness? The authors do mention the lack of calling rate differences between control HQ and LQ males and address some of its implications at the end of the discussion, but this finding should be stated in the results, and presented earlier and in a more nuanced manner in the discussion.
Furthermore, the lack of difference in condition complicates the authors attempt to assess how individual quality mediates the effect of an immune challenge on reproductive investment (lines 84-85) because the underlying assumption is that condition, which in turn is assumed to influence mate quality, represents available resources to be allocated into different fitness-enhancing traits/behaviours. It would be helpful if the authors address more directly the links between attractiveness and condition (which are sometimes represented as if they were a single trait: individual quality).

Minor comments
1) It would be helpful if early in the introduction (lines 103 to 112) the authors clarify that they used both control HQ and LQ males (this is not obvious until the methods).
2) The text is very well written but perhaps a figure with the predictions would be helpful.
3) There are a few typos - Line 41: “trait” instead of “traits”; Line 43 remove “that” or add “we found”; Line 73: should you add “and female preference”?; Line 127: remove “male”; Line 258: “strategy” instead of “strategy”.

---

## Round 0.2 · accepted · Accept

· Academic Editor

Accept

I appreciated your attention to detail in revising your manuscript. Given your revisions captured most of the issues raised by the reviewers, I am pleased to accept your manuscript for publication. I was pleased that even when you disagreed with the reviewers suggestions, you took strides to clarify your manuscript so that future readers would not have similar concerns.